# HANRAG: Heuristic Accurate Noise-resistant Retrieval-Augmented Generation for Multi-hop Question Answering

## Abstract

The Retrieval-Augmented Generation (RAG) approach enhances question-answering systems and dialogue generation tasks by integrating information retrieval (IR) technologies with large language models (LLMs). This strategy, which retrieves information from external knowledge bases to bolster the response capabilities of generative models, has achieved certain successes. However, current RAG methods still face numerous challenges when dealing with multi-hop queries. For instance, some approaches *overly rely on iterative retrieval*, wasting too many retrieval steps on compound queries. Additionally, using the original complex query for retrieval may fail to capture content relevant to specific sub-queries, resulting in *noisy retrieved content*. If the noise is not managed, it can lead to the problem of *noise accumulation*. **To address these issues, we introduce HANRAG**, a novel heuristic-based framework designed to efficiently tackle problems of varying complexity. Driven by a powerful revelator, HANRAG routes queries, decomposes them into sub-queries, and filters noise from retrieved documents. This enhances the system's adaptability and noise resistance, making it highly capable of handling diverse queries. We compare the proposed framework against other leading industry methods across various benchmarks. The results demonstrate that our framework obtains superior performance in both single-hop and multi-hop question-answering tasks. We will release our code and dataset in the future.

## 1 Introduction

Benefiting from the rapid development of large language models (LLMs) OpenAI et al. (2024); Grattafiori et al. (2024); Bai et al. (2023); Choi et al. (2024); DeepSeek-AI et al. (2024) in recent years, various natural language processing tasks, including text summarization Basyal & Sanghvi (2023); Fang et al. (2024) and query answering (QA) Patel et al. (2024); Balepur et al. (2025), have been effectively addressed. However, LLMs, which are trained on large amounts of text data, inevitably encounter some issues, such as outdated training data leading to problems with the timeliness of generated results Zhu et al. (2024), as well as factual inaccuracies due to errors in the training corpus. Huang et al. (2025)

To address the issue of generating incorrect results solely relying on the internal knowledge of LLMs, researchers have proposed Retrieval-Augmented Generation (RAG) techniques Gao et al. (2024); Fan et al. (2024); Edge et al. (2024); Yan et al. (2024). These techniques leverage information retrieval from external knowledge sources and integrate the retrieved information into prompts to help generate more accurate answers for queries. However, most of the current research focuses on single-hop queries, with relatively few studies addressing the more exploratory multi-hop queries. Among the few RAG methods designed to tackle multi-hop queries, there are still significant challenges.

**C1**: **Excessive dependence on iterative retrieval.** Most research on multi-hop queries tends to focus on solving complex queries Jiang et al. (2025); Shao et al.; Jeong et al. (2024); Trivedi et al. (2023); Gan et al. (2024), often overlooking a more common type of problem: **compound queries**. Compared to complex queries, compound queries typically seek answers to multiple aspects of a

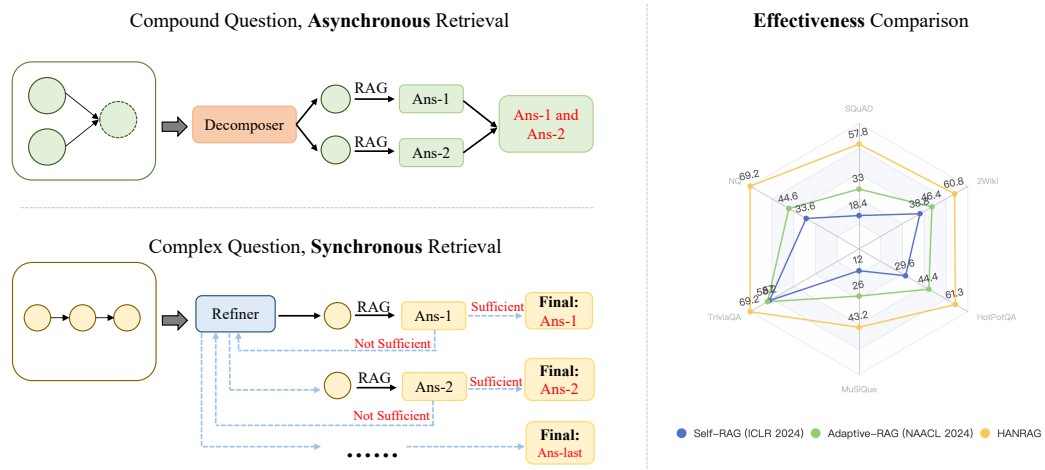

Figure 1: **Left.** Comparison of retrieval methods for Compound vs. Complex queries. Complex queries consist of interdependent sub-queries requiring strong logical reasoning, so synchronous retrieval is needed. In contrast, Compound queries have mostly independent sub-queries, making asynchronous retrieval more efficient. **Right.** Accuracy comparison of different methods across single and multi-hop benchmarks.

single subject, with the answer consisting of multiple facts. To the best of our knowledge, previous RAG systems that address multi-hop queries have almost all used iterative retrieval to solve compound multi-hop queries. This results in multiple rounds of alternating retrieval and generation, which is inherently inefficient.

**C2**: **Irrational querying.** Many methods use the original query, or the user input query, as the basis for multiple retrieval rounds. This often leads to difficulties in retrieving the relevant information needed to answer sub-queries for more complex queries, such as those with three or more hops, thereby weakening the system's overall performance when answering the original query. EfficientRAG Zhuang et al. (2024), while taking into account the review of outstanding questions during each retrieval, cannot accurately determine which issues need to be addressed next based solely on the original query and the refined chunk content.

**C3**: **Noise Accumulation.** The absence of post-processing operations on retrieved content can lead to large language models (LLMs) receiving noise information that is irrelevant to the original query. In iterative retrieval processes, each round typically necessitates the extraction of numerous documents, inevitably introducing extraneous noise that can significantly hinder the performance of subsequent LLMs. Therefore, without training the LLM, an effective information retrieval system must possess robust noise resilience. However, current approaches Xu et al.; Wang et al. predominantly employ overly fine-grained methods to filter content at the character or word level, which results in inefficiencies within the overall system operation.

To address the aforementioned challenges, this paper proposes a novel framework called **HAN-RAG**. At its core is a robust and versatile master agent named "Revelator," which orchestrates the process by routing diverse queries, decomposing compound problems, refining complex questions, and adaptively resolving real-world challenges. To tackle challenge **C1**, we introduce a previously unexplored pathway during query routing, leveraging parallel retrieval mechanisms to resolve compound questions more efficiently. This enhancement significantly improves the overall effectiveness of the RAG system. Furthermore, we constructed a benchmark composed of 2- to 4-hop compound problems to evaluate our system's performance in handling multi-step queries. For challenge **C2**, we implemented an iterative process wherein the system refines "seed questions"—the sub-questions derived from existing information that need to be answered at each step. These seed questions guide subsequent retrieval rounds, and the process continues iteratively until the original query is resolved. To address challenge **C3**, we leverage Revelator to evaluate the relevance of each retrieved document to the query. This enables us to filter out irrelevant content and pass only the most accurate and pertinent information to the LLM for answering each iteration's seed question.

In summary, our key contributions can be summarized as follows:

1. We propose a novel, **Revelator**-driven RAG framework with robust noise resistance and exceptional adaptive capabilities, referred to as **H**euristic **A**ccurate **N**oise-resistant **RAG (HANRAG)**.

2. We introduce a high-performance heuristic training method for the revelator, which results in a remarkable heuristic model that routes queries, and handles different types of multi-hop queries. Furthermore, we provide a benchmark comprising compound multi-hop queries.

3. Experimental results show that our proposed framework not only performs excellently on multi-hop benchmarks but also achieves impressive performance on single-hop benchmarks.

## 2 RELATED WORK

**Multi-hop Query Answering (QA)** is an open-domain query answering paradigm designed for complex reasoning tasks Mavi et al. (2024); Trivedi et al. (2019). Its core feature lies in requiring the query answering system to iteratively retrieve distributed knowledge sources through a multi-step reasoning chain, ultimately deriving entailed answers Jiang et al. (2025); Shao et al.. This raises higher performance demands for the QA system, which must continually interleave between retrieval, generation, and termination judgment processes. The system must also analyze implicit cross-document relationships to derive the correct answer to the original query. This multi-document answer retrieval approach overcomes the limitations of traditional single-hop QA, which is confined to a single document.

The typical workflow of multi-hop QA involves three core components: an embedding-based retriever, an answer generator, and a discriminator to determine whether further retrieval is necessary Ribeiro et al. (2022); Liu et al. (2024); Zhuang et al. (2024). The retriever's primary function is to retrieve relevant documents from an external knowledge base based on the query. The generator then uses the retrieved documents to formulate an answer to the query. The discriminator then assesses whether further retrieval is needed. If the original query has been sufficiently answered, no further retrieval is required; otherwise, additional retrieval is triggered.

Compared to single-hop QA systems Zhu et al. (2021); Chen et al. (2024); Nian et al. (2024), the key challenges in multi-hop QA include the accuracy of cross-domain information retrieval, the lack of strong reasoning capabilities in generation models, and the noise accumulation from multiple retrieval iterations.

**Iterative Retrieval-Augmented Generation (Iterative RAG)**, also known as Recursive Retrieval-Augmented Generation (Recursive RAG) Jiang et al. (2025); Shao et al., is an advanced mode of the traditional RAG framework, optimizing the final output through dynamic interaction between multiple rounds of retrieval and generation. The core of this method lies in expanding the single "retrieve-generate" cycle into a recursive process: after the initial generation of preliminary results, the system adjusts its retrieval strategy based on the generated content (such as rephrasing the original query, expanding the retrieval scope, or deciding whether to continue retrieving). This allows the system to acquire more precise contextual information from external knowledge bases and iteratively generate more refined answers. This approach is particularly effective for complex, multi-step queries (e.g., medical diagnoses, mathematical reasoning, etc.), addressing the fragmentation or information gaps that arise from the limitations of single-pass retrieval in traditional RAG. By allowing multiple rounds of self-correction, it significantly improves the coherence and factual accuracy of answers, making it a key technological path towards achieving human-like reasoning in intelligent query-answering, conversational systems, and expert tools.

**Adaptive Retrieval** is a method that dynamically adjusts the retrieval strategy based on the complexity of the user's query. Unlike traditional fixed retrieval models, it analyzes the user's query and automatically selects a retrieval path, optimizing the scope, depth, and granularity of the search. ADAPT-LLM Labruna et al. (2024) and Self-RAG Asai et al. (2023) introduce the idea of adding a special token to indicate that the LLM's internal knowledge alone cannot answer the user's query, necessitating the retrieval of external knowledge to assist in the response. This real-time retrieval-based approach aligns more closely with human thought processes. However, the delays caused by retrieval can significantly impact the efficiency of generating answers. Mallen et al. (2022) proposes a binary decision framework that labels queries based on the frequency of entities within the query

to determine whether retrieval is necessary. While this simple classification method works for basic queries, it struggles to handle more complex multi-hop tasks. To address this, finer-grained classification methods have been proposed. Jeong et al. (2024) introduces a framework called Adaptive-RAG, which classifies queries into categories such as straightforward queries, single-step queries, and multi-step queries, each with a different retrieval strategy. This approach adapts to all queries, but it shows notable inefficiency when dealing with compound queries.

## 3 PREAMBLE

In this section, we will provide detailed definitions of various types of problems and their corresponding solutions.

For **straightforward queries** that do not require retrieval, such as identity-related queries like *"Who are you?"*, or general knowledge queries like *"Who is the first President of America?"*, the LLM can directly answer using internal knowledge without external retrieval. Although external knowledge might enrich the answer, it does not affect its accuracy. Considering time efficiency, the optimal solution is for the LLM to answer the query directly.

For **single-step queries**, such as *"When was Pan Jianwei born?"*, LLMs often provide incorrect answers due to incomplete internal knowledge, a phenomenon known as "hallucination." To address such queries, external knowledge must be retrieved to ensure the accuracy of the generated response. Traditional RAG techniques typically use a single-step retrieval to answer such queries, as described in B.1.

**Compound queries**, such as *"When is the birthdate of Michael F. Phelps? When did he retire?"* typically ask about multiple attributes of a single entity. This implies that the answers to these queries contain multiple pieces of information, which may need to be addressed separately. When answering such queries, the query is often decomposed, and each sub-query is treated as an individual retrieval task. The corresponding formula is as follows:

$$A^* = \underset{A=\{a_1,\ldots,a_n\}}{\arg\max} \prod_{i=1}^{n} P_{\text{LLM}}(a_i \mid q_i; D) \quad \text{s.t. } \{q_1,\ldots,q_n\} = Decomposer(Q) \tag{1}$$

where $Q$ denotes the input compound query, $Decomposer(Q) = \{q_1,\ldots,q_n\}$ is its decomposition into $n$ sub-queries, $D$ is the retrieval knowledge base, and $A^* = \{a_1,\ldots,a_n\}$ the optimal answer set that maximizes the joint probability of all sub-query answers.

The issue of **complex queries** arises in queries like *"Who succeeded the first President of Namibia?"*. Those queries involve closely connected sub-queries that require complex logical reasoning to derive the right answer. To answer such queries, one must first refine the seed query, and after answering the first seed query, rephrase the remaining queries. Then, the next seed query is refined from the remaining ones, and this iterative process of alternating between retrieval and generation continues until the final answer is obtained. The formula can be expressed as follows:

$$A^* = \{a_1^*,\ldots,a_n^*\} = \arg\max_A \prod_{t=1}^{n} P_{\text{LLM}}(a_t|q_t, a_{1:t-1}; D) \quad \text{s.t. } q_t = \text{Refiner}_t(Q, a_{1:t-1}) \tag{2}$$

where $Q$ denotes the input complex query, $q_t$ is the refined sub-query at step $t$, and $a_t$ is the corresponding answer. $\text{Refiner}_t(Q, a_{1:t-1})$ represents the refinement function that reformulates the next sub-query based on the original query $Q$ and the previously obtained answers $a_{1:t-1}$. $D$ denotes the knowledge base used for retrieval, and $A^* = \{a_1^*,\ldots,a_n^*\}$ is the optimal trajectory of answers obtained by maximizing the joint probability across all reasoning steps.

## 4 METHODOLOGY

In this section, we provide a detailed introduction to the proposed general RAG framework, HAN-RAG. This framework demonstrates impressive adaptive capabilities across diverse query patterns.

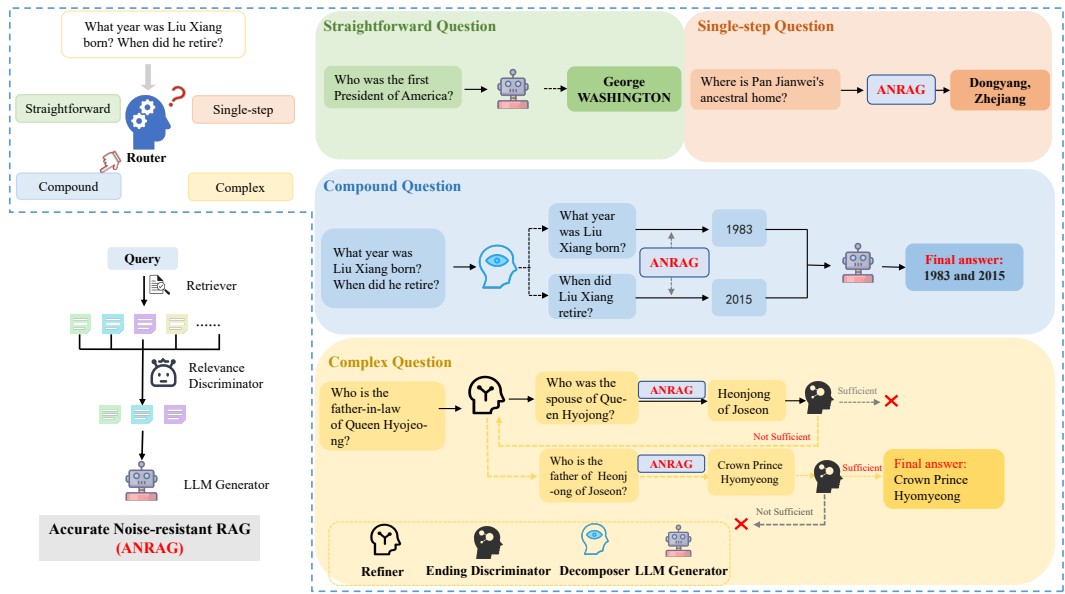

Figure 2: Overall framework of HANRAG. The top-left section illustrates the functioning of the router, routing the query to the correct category. The bottom-left section outlines the workflow of ANRAG, which determines the relevance between the document and the query, ensuring that only noise-free documents are passed to the generation model. The right-hand section provides a detailed depiction of how HANRAG handles four different types of queries: Straightforward queries are directly answered using the LLM; for Single-step queries, ANRAG is employed to derive the answer; Compound queries are addressed using asynchronous retrieval followed by result merging; and for Complex queries, synchronous retrieval is utilized, alternating between retrieval, generation, and termination evaluation to produce the final outcome. **All functions, apart from the LLM and the Retriever, are managed by the Revelator module.**

---

**Algorithm 1** ANRAG Framework

**Input:** Query $Q$, Passage collection $\mathcal{D}$, **Retriever**, **Revelator**$_{rel}$, **LLM**
**Output:** Top-3 Relevant Passages $\mathcal{D}_{rel}$
1: $\mathcal{D}_{top10} \leftarrow$ RETRIEVER$(Q, \mathcal{D})$
2: $\mathcal{D}_{rel} \leftarrow$ REVELATOR_REL$(Q, \mathcal{D}_{top10})$
3: **return** LLM$(\mathcal{D}_{rel})$ =0

---

## 4.1 FRAMEWORK

The overall framework of HANRAG is illustrated in Figure 2. It leverages a multifunctional master agent, dubbed "**Revelator**," to guide the entire framework in performing precise routing and retrieval, thereby inspiring terminal-level LLMs (Large Language Models) to generate more accurate responses.

Firstly, the HANRAG framework incorporates a noise-resistant one-step retrieval method named ANRAG, whose workflow is presented in the bottom left of Figure 2, with its algorithmic process depicted in Algorithm 1. Initially, ANRAG employs a **Retriever** to gather multiple passages relevant to the input query. Then, it uses the **Revelator** to assess the relevance between retrieved documents and input query. After filtering out unrelated documents, the remaining documents are passed to the **LLM** to produce final answer.

In the face of more dynamic and complex real-world scenarios, the **Revelator** first routes the query to the appropriate processing chain. Straightforward questions are routed directly to the LLM for quick response generation. For one-step retrieval questions, the query is directed to ANRAG for retrieval and answering.

For compound questions, the **Revelator** begins by breaking them down into multiple independent sub-questions. Each sub-question is treated as a single-step retrieval task, which can be effectively resolved through asynchronous execution of the single-step retrieval chain. The results of all sub-questions are then aggregated to form the answer to the original compound question.

For complex questions, the **Revelator** orchestrates an iterative retrieval process. It firstly refines a "seed question" from the complex reasoning task—essentially the first sub-question that needs to be answered, as its response is a prerequisite for addressing subsequent sub-questions. After deriving the seed question's answer through ANRAG, the **Revelator** evaluates whether the answer provides sufficient information to resolve the original complex question. If sufficient, the iterative process halts. If not, further iterations are performed until the solution to the original question is achieved.

In summary, the **Revelator** serves as the master agent that orchestrates and drives the entire framework with remarkable efficiency and precision. At the outset, it performs adaptive query routing, intelligently directing user queries to the most appropriate processing chain based on the nature and complexity of the request. Equipped with a built-in ability to decompose compound questions into manageable components and refine critical sub-questions, the **Revelator** can ensure that each query receives targeted and context-aware processing. After retrieving relevant information, the **Revelator** further filter out noise, ensuring that only the most pertinent documents are passed along for further processing. By combining adaptive query handling, sophisticated decomposition techniques, and rigorous post-retrieval refinement, the **Revelator** plays an indispensable role in this framework.

## 4.2 DATA CONSTRUCTION FOR REVELATOR

To enhance the performance of the entire framework, we have constructed a series of high-quality datasets to train the various capabilities of the **Revelator**. For **Routing**, the data format is $<Q, CLS>$. We collect four types of queries as training data. For *straightforward queries,* we use 9,741 samples from Talmor et al. (2019).

For *single-step retrieval queries*, we source data from two different origins: the first part comes from training data in single-step QA datasets like Natural Questions Kwiatkowski et al. (2019), and the second part comes from multi-hop QA datasets like Musique Trivedi et al. (2022), which contain all sub-queries that make up the complex query. We sample 50,000 from that. For *multi-hop complex queries*, we directly sample 50,000 data from the MuSiQue training data. For *multi-hop compound queries*, the data construction process can be found in C.2

---

**Algorithm 2** HANRAG Framework

---

**Input:** Query $Q$, Passage collection $D$, Revelator, LLM
**Output:** Final Answer $\hat{y}$
1: $cls \leftarrow \text{REVELATOR}(Q)$
2: **if** $cls = straightforward$ **then**
3:    **return** LLM$(Q)$
4: **else if** $cls = single$ **then**
5:    **return** ANRAG$(Q, D)$
6: **else if** $cls = compound$ **then**
7:    $\{q_1, q_2, \ldots, q_n\} \leftarrow \text{REVELATOR}(Q)$
8:    **for** $i = 1 \rightarrow n$ **do**
9:      $\hat{y}_i \leftarrow \text{ANRAG}(q_i, D)$
10:    **end for**
11:    **return** LLM$(Q, \{q_i, \hat{y}_i\}_{i=1}^n)$
12: **else if** $cls = complex$ **then**
13:    $is\_ending \leftarrow \text{REVELATOR}(Q, \emptyset)$
14:    **while** $is\_ending = No$ **do**
15:      $q_i \leftarrow \text{REVELATOR}(Q, \{q_t, \hat{y}_t\}_{t<i})$
16:      $\hat{y}_i \leftarrow \text{ANRAG}(q_i, D)$
17:      $is\_ending \leftarrow \text{REVELATOR}(Q, \{q_t, \hat{y}_t\}_{t \leq i})$
18:    **end while**
19:    **return** LLM$(Q, \{q_i, \hat{y}_i\}_{i=1}^k)$
20: **end if**=0

---

For **Decomposition**, the data format is $<Q, q_1, q_2,... >$. we use the aforementioned multi-hop compound queries and their sub-queries directly as training data.

For **Refinement**, the data format is $<Q, q_i >$. we utilize the detailed reasoning processes in the MuSiQue Trivedi et al. (2022) and 2Wiki Ho et al. (2020) datasets. Each reasoning step serve as a seed query for training the refiner.

For **Relevance Discrimination**, we collect $<Q, D, IS\_REL >$ sample pairs as training data. Here, $<Q, D >$ includes queries from single-hop benchmarks along with their corresponding corpora, as well as sub-queries from complex multi-hop benchmarks and their corpora. Using Qwen2-72B-instruct for relevance annotation, we can obtain $<Q, D, Rel >$ pairs.

For **Ending Discrimination**, the data format is $<Q, q_1, \hat{y}_1,..., IS\_ENDING >$, we similarly synthesize using the reasoning processes contained in the MuSiQue and 2Wiki datasets. We consider the sub-query and answer from the last hop as the basis for determining the completion of the original query, meaning no further retrieval is needed. Conversely, the sub-queries and answers from previous hops are used as an indication that the original query is incomplete, necessitating further retrieval. This approach is used to synthesize the training data for the Ending discriminator.

We adopt a multi-task **mixed-training strategy** for training Revelator, which is simpler and more straightforward to execute than training each task separately. Additionally, it is important to note that there is **no overlap** between our training and testing data. This ensures the validity and authenticity of the evaluation.

# 5 EXPERIMENTS

**Datasets.** To validate the effectiveness and efficiency of the method we proposed, we conducted evaluations on a variety of benchmarks, including three single-hop query datasets, three multi-hop complex query datasets, and one multi-hop query compound dataset that we introduced.

For the single-hop query datasets, we used the same benchmarks as Jeong et al. (2024), including SQuAD v1.1 Rajpurkar et al. (2016), Natural Questions Kwiatkowski et al. (2019), and TriviaQA Joshi et al. (2017).

For the multi-hop query datasets, we followed the configuration from Jeong et al. (2024), which includes HotpotQA Yang et al. (2018), 2WikiMultihopQA Ho et al. (2020), and MuSiQue Trivedi et al. (2022).

A detailed description of the construction process of the compound benchmark can be found in the supplementary materials C.1.

**Methods.** We conduct a comprehensive comparison of the proposed HANRAG with various methods, including 1) a method that directly generates answers via Large Language Models (LLMs) without retrieval; 2) a single-step retrieval method designed for addressing single-hop queries, specifically utilizing BM25 Robertson et al. (1994) as the retriever to obtain the top 5 documents relevant to the query to assist LLMs in generating answers; 3) an iterative retrieval method IRCoT Trivedi et al. (2023) for tackling complex queries; and 4) several adaptive methods such as Adaptive Retrieval Mallen et al. (2022), self-RAG Asai et al. (2023), and Adaptive-RAG Jeong et al. (2024). Additionally, we also align our setup with Adaptive-RAG by using FLAN-T5-XL (3B) Raffel et al. (2019) for both the Revelator and LLM generator, conducting an additional experiment to ensure fairness, named **HANRAG-Fair**. The goal is to demonstrate the versatility and generalizability of the proposed method. The ablation study can be found in F.

**Metrics.** We primarily evaluate two aspects: effectiveness and efficiency. For **effectiveness**, we leverage EM, F1, and accuracy (Acc) as evaluation metrics. EM indicates whether the ground truth is exactly equal to the prediction, F1 measures the overlap of words between the ground truth and prediction, and Acc denotes whether the ground truth is included in the prediction.

For evaluating **efficiency**, we use the number of steps as a metric, defined as the number of retrieval-generation cycles required to resolve a given query.

**Implementation Details.** For both single-hop and multi-hop benchmarks, we follow the processing approach outlined in Jeong et al. (2024) to create the training, development, and test datasets. We

apply BM25 as the Retriever. For the Revelator and LLM generator, we use the llama-3.1-8b-instruct model Grattafiori et al. (2024). The Revelator requires fine-tuning due to the complexity of its tasks, whereas the LLM generator does not need additional training. Training configurations can be found in E.

# 6 RESULTS

The experimental results show that HANRAG achieves efficiency and effectiveness for all types of queries, especially in terms of accuracy (Figure 3). The specific analysis is as follows.

**Effect and Efficiency of single-hop dataset.** The performance of HANRAG and all comparative methods on the **single-hop dataset** is presented in Table 1. As can be observed, HANRAG achieves optimal results across all benchmark evaluation metrics. Specifically, HANRAG outperforms Adaptive-RAG by margins of 12.2%, 6.83%, and 20.13% on the EM, F1, and Accuracy metrics, respectively. This advantage can be attributed to the ability of the Revelator to filter out a significant portion of noisy retrievals. The presence of noise in retrieved content is a common challenge faced by vector-based retrieval methods or BM25. However, the Revelator, equipped with a strong semantic understanding capability, can effectively evaluate the relevance between each document and the original query, ensuring that more accurate documents are passed to the generation model. The results of the relevance discriminator ablation experiments and detailed case studies further support this conclusion. Additionally, the steps metric—representing the average number of retrieval steps—can be reduced by approximately 0.13 on average across the three datasets as shown in Figure 3. This improvement stems from the Revelator's robust adaptability, which allows it to precisely route single-hop questions to single-step retrieval pathways, thereby avoiding unnecessary multi-step retrieval processes.

Table 1: Results on single-hop benchmark. **Bold** and underlined text indicate the optimal and sub-optimal results (excluding "Steps" in "No retrieval").

| Methods | SQuAD | | | | Natural Questions | | | | TriviaQA | | | |
|---|---|---|---|---|---|---|---|---|---|---|---|---|
| | EM | F1 | Acc | Steps | EM | F1 | Acc | Steps | EM | F1 | Acc | Steps |
| **No retrieval** | 3.60 | 10.50 | 5.00 | 0.00 | 14.20 | 19.00 | 15.60 | 0.00 | 25.00 | 31.80 | 27.00 | 0.00 |
| **Single-step Approach** | 27.80 | 39.30 | 34.00 | 1.00 | 37.80 | 47.30 | 44.60 | 1.00 | 53.60 | 62.40 | 60.20 | 1.00 |
| **IRCoT** (ACL 2023) | 24.40 | 35.60 | 29.60 | 4.52 | 38.60 | 47.80 | 44.20 | 5.04 | 53.80 | **62.40** | 60.20 | 5.28 |
| **Adaptive Retrieval** (ACL 2023) | 13.40 | 23.10 | 17.60 | **0.50** | 28.20 | 36.00 | 33.00 | **0.50** | 38.40 | 46.90 | 42.60 | **0.50** |
| **Self-RAG** (ICLR 2024) | 2.20 | 11.20 | 18.40 | 0.63 | 31.40 | 39.00 | 33.60 | 0.63 | 12.80 | 29.30 | 57.00 | 0.63 |
| **Adaptive-RAG** (NAACL 2024) | 26.80 | 38.30 | 33.00 | 1.37 | 37.80 | 47.30 | 44.60 | 1.00 | 52.20 | 60.70 | 58.20 | 1.23 |
| **HANRAG** | **39.80** | **39.76** | **57.80** | 1.11 | **56.40** | **49.12** | **69.20** | 1.00 | **57.20** | 62.21 | **69.20** | 1.08 |
| **HANRAG-Fair** | 32.60 | 44.90 | 53.80 | 1.13 | 50.50 | 54.40 | 64.80 | 1.06 | 52.40 | 60.90 | 63.60 | 1.11 |

Table 2: Results on multi-hop benchmark. **Bold** and underlined text indicate the optimal and suboptimal results (excluding "Steps" in "No retrieval" and "single-step" methods).

| Methods | Musique | | | | HotpotQA | | | | 2WikiMultihopQA | | | | CompoundMultihop | |
|---|---|---|---|---|---|---|---|---|---|---|---|---|---|---|
| | EM | F1 | Acc | Steps | EM | F1 | Acc | Steps | EM | F1 | Acc | Steps | Acc | Steps |
| **No retrieval** | 2.40 | 10.70 | 3.20 | 0.00 | 16.60 | 22.71 | 17.20 | 0.00 | 27.40 | 32.04 | 27.80 | 0.00 | – | – |
| **Single-step Approach** | 13.80 | 22.80 | 15.20 | 1.00 | 34.40 | 46.15 | 36.40 | 1.00 | 41.60 | 47.90 | 42.80 | 1.00 | – | – |
| **IRCoT** (ACL 2023) | 23.00 | 31.90 | 25.80 | 3.60 | 44.60 | 56.54 | 47.00 | 5.53 | **49.60** | 58.85 | 55.40 | 4.17 | – | – |
| **Adaptive Retrieval** (ACL 2023) | 6.40 | 15.80 | 8.00 | **0.50** | 23.60 | 32.22 | 25.00 | **0.50** | 33.20 | 39.44 | 34.20 | **0.50** | – | – |
| **Self-RAG** (ICLR 2024) | 1.60 | 8.10 | 12.00 | 0.73 | 6.80 | 17.53 | 29.60 | 0.73 | 4.60 | 19.59 | 38.80 | 0.73 | – | – |
| **Adaptive-RAG** (NAACL 2024) | 23.60 | 31.80 | 26.00 | 3.22 | 42.00 | 53.82 | 44.40 | 3.55 | 40.60 | 49.75 | 46.40 | 2.63 | 52.13 | 2.76 |
| **HANRAG** | **29.80** | **36.60** | **43.20** | 2.45 | **49.20** | **58.90** | **61.30** | 3.07 | 47.20 | **58.90** | **60.80** | 2.31 | **71.76** | **1.24** |
| **HANRAG-Fair** | 26.80 | 34.10 | 39.20 | 2.86 | 46.90 | 56.40 | 58.80 | 3.19 | 45.40 | 55.30 | 57.60 | 2.39 | 64.30 | 1.68 |

**Effect and efficiency of complex queries**. HANRAG also demonstrates exceptional performance, as shown in Table 2. Across the three complex-problem datasets, HANRAG consistently outperforms Adaptive-RAG on the EM, F1, and Accuracy metrics, with average improvements of 6.67%, 6.34%, and 16.17%, respectively. This performance gain is attributed to the guidance role provided by the Revelator throughout the iterative retrieval process. The Revelator effectively identifies the

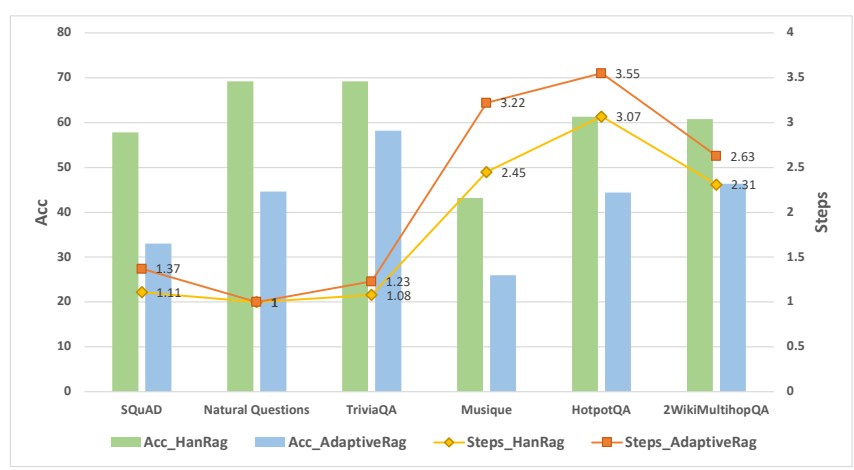

Figure 3: Comparison of retrieval methods for Single and Complex queries.

next sub-question that needs to be addressed and, after thoroughly understanding the documents retrieved by the Retriever, retains only those that are relevant to answering the question. This significantly minimizes the interference of noisy documents in the overall generation process, preventing the accumulation of noise over multiple retrieval steps. For efficiency, compared to Adaptive-RAG, HANRAG reduces the average number of retrieval steps by 0.52, saving both time and computational resources for each complex problem as shown in Figure 3. This improvement is attributed to the Revelator's precise judgment after each retrieval-generation cycle, accurately deciding whether further retrieval is necessary. Such an efficient approach underscores HANRAG's potential for broader applicability in real-world scenarios.

**Effect and efficiency of multi-hop compound queries**, we compare our proposed HANRAG with Adaptive-RAG on the compound query dataset we have built. Since the ground truth for such queries involves multiple entities, we use Accuracy and Steps as evaluation metrics for effectiveness and efficiency, respectively. The calculation method involves splitting the ground truth of each query into multiple entities, treating each entity present in the prediction as a positive sample, and calculating accuracy accordingly. The results are shown in Table 2. It is evident that HANRAG exceeds Adaptive-RAG in accuracy by 19.63%, and HANRAG shows a significant reduction of steps by nearly 1.5 compared to Adaptive-RAG. These results suggest that HANRAG demonstrates strong performance in both decomposing and answering sub-queries, while efficiently resolving compound queries without the need for excessive iterative retrieval.

**Ablation Study**. Furthermore, the ablation study on the various functionalities of Revelator demonstrates that the performance of HANRAG decreases when any single module is removed while solving complex problems. Notably, the absence of the Refiner leads to a significant drop of 10% in overall accuracy, highlighting the importance of extracting seed problems in addressing complex issues. For detailed results, please refer to the appendix F.

## 7 CONCLUSION

In this paper, we provide a more detailed classification of multi-hop queries, including compound and complex queries, and propose a highly adaptive RAG framework, HANRAG. This framework leverages a "Master Agent" called the **Revelator**, which as a **Router** precisely routes any query to the corresponding chain of reasoning for targeted resolution. After accurately routing all types of queries, we use the **Revelator as a Decomposer** to decompose compound queries into sub-queries, then parallelly retrieve and generate answers for each sub-query, ultimately aggregating them to form the final answer. For complex queries, the **Revelator is employed as an refiner** to refine seed queries, followed by retrieval and generation of answers for these seed queries. Through an alternating process of seed query refinement, retrieval, generation, and ending judgment, we progressively arrive at the final solution. Additionally, we introduce a high-performance **Relevance discrimina-**

**tor**, designed to assess the relevance of documents retrieved by the Retriever to the query at hand, thereby enabling effective filtering of noise documents. By constructing high-quality training data and employing multi-task learning, the functionalities of the above modules are integrated into Revelator, we ensure that HANRAG outperforms several SOTA methods in comparative evaluations, establishing itself as the latest SOTA approach.

## ETHICS STATEMENT

Our research utilizes LLMs, and we acknowledge that these models may inherit biases present in their training data. We evaluate our framework on public benchmarks to ensure fair and reproducible comparisons. We recognize the potential for LLMs to generate inaccurate or harmful content, and believe our approach—which improves retrieval accuracy by filtering noise—is a step toward building more reliable and responsible AI systems

## REPRODICIBILITY STATEMENT

To support the reproducibility of our findings, we will publicly release all source code, datasets, and model weights in a GitHub repository upon the paper's acceptance. The repository will include detailed instructions for replicating our experiments.

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

## A    USE OF LLM

In conducting this study, we leveraged LLMs, which were primarily employed to retrieve and summarize prior research. This significantly streamlined our research process and enhanced overall efficiency.

## B    FORMULAS

### B.1    SINGLE-STEP RETRIEVAL

The formula for Single-step Retrieval as follows:

$$A^* = \arg\max_A P_{\text{LLM}}(A|Q, D) \quad \text{s.t. } D_k = \text{Retriever}(Q, D) \tag{3}$$

Here, $Q$ represents the user's input query; $D$ denotes the external knowledge base, which is typically composed of authoritative documents; $Retriever$ is a model used to retrieve relevant information from the external knowledge base, often implemented by embedding models; $D_k$ represents the fixed number of documents selected from retrieved documents, which are then input into the LLM; $A^*$ indicates the result generated by the LLM.

### B.2    ENDING DISCRIMINATOR

The formula for Ending discriminator as follows:

$$is\_ending = discriminator_{end}(\mathcal{Q}, q_1, \hat{y}_1, q_2, \hat{y}_2, ...) \tag{4}$$

$is\_ending$ indicates whether the iterative process has ended. A value of 0 means that the iterative retrieval-generation process will continue, while a value of 1 means that the current reasoning step is sufficient to answer the original complex query, and no further retrieval-generation is needed.

Q represents the original complex reasoning query, $q_k$ represents the k-th seed query generated during the retrieval process, and $\hat{y}_k$ represents the predicted answer to the sub-query $q_k$.

### B.3    RELEVANCE DISCRIMINATOR

The formula for Relevance discriminator as follows:

$$is\_rel = discriminator_{rel}(q, d), is\_rel \in \{0, 1\} \tag{5}$$

$is\_rel$ indicates whether the query q is relevant to the retrieved document d. A value of 0 means the document is irrelevant, while a value of 1 means the document is relevant. Relevant documents are retained for downstream processing, while irrelevant documents are discarded.

## C    DETAILS OF DATA CONSTRUCTION

### C.1    COMPOUND QUERIES BENCHMARK

We start by randomly selecting 10,000 entities from the Wikipedia corpus. For each entity, we extract 10 corresponding documents to form <ENTITY, DOC >sample pairs. Using the Prompt 15, we request Qwen2-72B-instruct Yang et al. (2024) to generate a question based on each document, ensuring that the question focuses on the given entity and that the answer can be found within the provided DOC. This process results in <ENTITY, DOC, q, a >sample pairs. Then, we randomly combine queries via Prompt 16 for the same entity to create 2-4 hop compound questions and their corresponding answers <ENTITY, DOCS, Q, A >, from which we sample 50,000 entries as training data, 8000 samples for dev and 2000 for test.

## C.2 ROUTER

For **Router**, the data format is $<Q, CLS>$. We collect four types of queries as training data.

For straightforward queries, we use 12,247 samples from Talmor et al. (2019).

For single-step retrieval queries, we source data from two different origins: the first part comes from training data in single-step QA datasets like Natural Questions Kwiatkowski et al. (2019), and the second part comes from multi-hop QA datasets like Musique Trivedi et al. (2022), which contain all sub-queries that make up the complex query. We sample 50,000 from that.

For multi-hop complex queries, we directly sample 50,000 data from the Musique training data.

For multi-hop compound queries, we extract the queries from the training data in the compound benchmark C.1 to train the Router for directing data to the compound category.

Table 3: Additional results on single-hop benchmark.

| Methods | SQuAD | | | | Natural Questions | | | | TriviaQA | | | |
|---|---|---|---|---|---|---|---|---|---|---|---|---|
| | EM | F1 | Acc | Steps | EM | F1 | Acc | Steps | EM | F1 | Acc | Steps |
| **Adaptive-RAG** (NAACL 2024) | 26.80 | 38.30 | 33.00 | 1.37 | 37.80 | 47.30 | 44.60 | 1.00 | 52.20 | 60.70 | 58.20 | 1.23 |
| **HANRAG** | 39.80 | 39.76 | 57.80 | 1.11 | 56.40 | 49.12 | 69.20 | 1.00 | 57.20 | 62.21 | 69.20 | 1.08 |
| **HANRAG-Oracle** | 41.7 | 52.2 | 59.3 | 1.0 | 56.8 | 61.2 | 70.7 | 1.0 | 58.4 | 64.5 | 71.6 | 1.0 |

Table 4: Additional results on multi-hop benchmark.

| Methods | Musique | | | | HotpotQA | | | | 2WikiMultihopQA | | | |
|---|---|---|---|---|---|---|---|---|---|---|---|---|
| | EM | F1 | Acc | Steps | EM | F1 | Acc | Steps | EM | F1 | Acc | Steps |
| **Adaptive-RAG** (NAACL 2024) | 23.60 | 31.80 | 26.00 | 3.22 | 42.00 | 53.82 | 44.40 | 3.55 | 40.60 | 49.75 | 46.40 | 2.63 |
| **HANRAG** | 29.8 | 36.6 | 43.2 | 2.45 | 49.2 | 58.9 | 61.3 | 3.07 | 47.2 | 58.9 | 60.8 | 2.31 |
| **HANRAG-Oracle** | 31.3 | 38.2 | 45.7 | 2.37 | 52.9 | 61.3 | 65.8 | 2.91 | 49.8 | 62.3 | 63.6 | 2.24 |

## C.3 DECOMPOSER

For the **Decomposer**, we directly use the aforementioned multi-step parallel retrieval problems as training data. Specifically, we extract the training samples $<$ENTITY, DOCS, Q, A$>$ from the compound benchmark, where $Q$ is the compound query. We then retrieve the multiple sub-queries that constitute $Q$ from storage, forming training samples in the format $<Q, q_1, q_2, \ldots>$

## C.4 REFINER

For the **refiner**, the MuSiQue and 2Wiki datasets contain comprehensive reasoning processes. We can use these step-by-step reasoning processes as seed problems for training the refiner. Additionally, to address the issue of single-step problems being incorrectly routed into a multi-step serial process, we also include single-step problems as both input and output in our training data.

## D ADDITIONAL EXPERIMENTAL RESULTS

**Router Performance.** We constructed a test set for the Router to evaluate its performance. The test set was built as follows: 1) We merged the dev and test splits of CommonSenseQA, then randomly sampled 1,500 queries to create the straightforward question set. 2) Queries from the three single-step retrieval test sets were combined to form the single-step question set, resulting in 1,500 queries. 3) Queries from the three complex problem test sets were merged to form the complex question set, also totaling 1,500 queries. 4) Lastly, 1,500 queries were randomly sampled from the compound problem test set to create the compound question set. Using the Revelator, we categorized the test set into these four distinct types of questions, achieving an accuracy of **83.93%**.

In addition to assessing the overall performance of our framework, we also evaluated HANRAG under ideal conditions — specifically, assuming the Router achieves 100% accuracy. We refer to

this variant as HANRAG-Oracle, and the detailed results are presented in the table 3. Compared to HANRAG, HANRAG-Oracle shows minor improvements in the three single-step retrieval test sets, with increases of 1.17% in EM, 1.8% in F1, and 1.8% in Accuracy. The small gains can be attributed to the fact that routing errors contribute minimally to incorrect answers in single-step retrieval problems. Most errors here stem from retrieval inaccuracies rather than routing missteps. Regarding the average steps metric, HANRAG-Oracle reduces the steps by only 0.06, primarily because the original Revelator already effectively classified test data without requiring ideal conditions.

For the three complex benchmarks, the results are shown in the table 4. HANRAG-Oracle achieves slightly higher improvements, increasing EM by 2.6%, F1 by 2.47%, and Accuracy by 3.27%. However, the gains remain limited, mainly due to the generative model's challenges in following instructions precisely and its insufficient ability to locate answers within a large scope ("needle-in-a-haystack" scenarios). On the average steps metric, HANRAG-Oracle improves by only 0.1. This marginal improvement is attributed to the already-strong routing capabilities of the original Revelator; further enhancement results in diminishing returns due to the effective performance of the baseline.

On the compound benchmark, shown as teble 5, HANRAG-Oracle achieves only a modest improvement in Accuracy compared to HANRAG, with an increase of just 1.36%. Most of the errors were caused by misaligned answer formats in the generated responses, meaning that even enhancing the Revelator's routing capabilities could not effectively resolve these issues. Regarding the steps metric, since all queries were accurately classified as compound problems, only a single retrieval step was needed for each question. As a result, the number of retrieval steps was reduced by 0.24 compared to HANRAG. While HANRAG-Oracle did not deliver significant improvements in accuracy, it optimized retrieval efficiency to its best possible level.

Table 5: Results on compound multi-hop benchmark

| Methods | CompoundMultihop | |
| --- | --- | --- |
| | Acc | Steps |
| Adaptive-RAG | 52.13 | 2.76 |
| HANRAG (ours) | 71.76 | 1.24 |
| HANRAG-Oracle | 73.12 | 1.0 |

# E   IMPLEMENTATION DETAILS

We utilized 8 Nvidia A100 GPUs and trained our model using the LLama-Factory framework Zheng et al. (2024). To enhance the efficiency of the training process, we applied the LoRA Hu et al. (2021) method for fine-tuning, training for only one epoch. The initial learning rate was set to 1.0e-4, with the scheduler type configured as cosine. Additionally, 10% of the training samples were used for a warm-up phase to ensure a smooth start.

# F   ABLATION STUDY

To clearly demonstrate the role of each component in our proposed HARAG, we conduct ablation experiments on the MuSiQue dataset. The experimental results are shown in the table below: A "-" indicates the absence of a particular module. We can observe a decline in performance when the Relevance discriminator module is missing. After examining the reasoning process of the pipeline, we find that much of this performance drop is due to noise, directly confirming the effectiveness of our proposed Relevance discriminator. When the Ending discriminator is absent, there is virtually no noticeable change in the overall framework's performance; however, the number of Steps increases to 4.5, which is the maximum retrieval step limit we set as a hyperparameter. Upon reviewing the reasoning process, we discover that without the Ending discriminator, the pipeline continues retrieval-generation cycles even after obtaining the final answer. When the refiner is removed, we use the original query directly for each retrieval round. The results show that using the original query leads to inaccurate retrievals, ultimately misleading the LLM to generate incorrect content.

Table 6: Ablation study on MuSiQue

| Methods | MuSiQue | | | |
|---------|-----|-----|-----|-------|
| | EM | F1 | Acc | Steps |
| Adaptive-RAG | 23.60 | 31.80 | 26.00 | 3.22 |
| HANRAG (ours) | 29.8 | 36.6 | 43.2 | 3.01 |
| -Relevance discriminator | 25.2 | 32.5 | 37.8 | 3.06 |
| -Ending discriminator | 28.7 | 35.8 | 44.3 | 4.56 |
| -Refiner | 24.2 | 32.4 | 28.9 | 4.12 |

## G  CASE STUDY

**Query:**  Which English King was married to Edith Swan-Neck, also known as Edith the Fair?
**Groundtruth:**  Harold II

**Adaptive-RAG:**
**Query Type:** B ( Single-step Query)
**Retrieved Doc1:** Edith of Wessex: brother-in-law. Edith was originally named Gytha, but re-named Ealdgyth (or Edith) when she married King Edward the Confessor. Her brothers were Sweyn (c. 1020 – 1052), Harold (later Harold II) (c. 1022 – 1066), Tostig (c. 1026 – 1066),... but Sweyn was the firstborn and Harold was the second son.
**Retrieved Doc2:** Edith Pargeter: Edith Pargeter Edith Mary Pargeter, OBE, BEM (28 September 1913 – 14 October 1995), also known by her ñom de plumeËllis Peters,... She was educated at Dawley Church of England School and
**Retrieved Doc3:** Edith Rigby: Edith Rigby Edith Rigby (ñéeÄayner) (18 October 1872 – 1948) was an English suffragette and ... She married Dr. Charles Rigby and lived
**Input to LLM Generator:** Doc1, Doc2, Doc3
**LLM Prediction:** Edith Mary Pargeter

**HANRAG:**
**Query Type:** B ( Single-step Query)
**Retrieved Doc1:** Edith of Wessex: brother-in-law. Edith was originally named Gytha, but re-named Ealdgyth (or Edith) when she married King Edward the Confessor. Her brothers were Sweyn (c. 1020 – 1052), Harold (later Harold II) (c. 1022 – 1066), Tostig (c. 1026 – 1066),... but Sweyn was the firstborn and Harold was the second son.
**Retrieved Doc2:** Edith Pargeter: Edith Pargeter Edith Mary Pargeter, OBE, BEM (28 September 1913 – 14 October 1995), also known by her ñom de plumeËllis Peters,... She was educated at Dawley Church of England School and
**Retrieved Doc3:** Edith Rigby: Edith Rigby Edith Rigby (ñéeÄayner) (18 October 1872 – 1948) was an English suffragette and ... She married Dr. Charles Rigby and lived
**Input to LLM Generator:** Doc1
**LLM Prediction:** Harold II

Table 7: Comparison between HANRAG and Adaptive-RAG for Single-Step Query. Under the same retrieval query and retriever, both Adaptive-RAG and HANRAG retrieve the same three documents. However, some of these documents, such as Doc2 and Doc3, are noisy. Adaptive-RAG passes these noisy documents to the LLM generator, which results in an incorrect final answer. In contrast, HANRAG effectively filters out the noisy documents (Doc2 and Doc3), preventing them from interfering with the LLM generator and enabling it to produce the correct answer. **To simplify the presentation, only 3 documents are displayed here.**

---

**Query:** When did Lionel Cranfield, 3rd Earl of Middlesex succeed his brother James as Earl of Middlesex and who is his wife?
**Groundtruth:** 1651 && Rachael

**Adaptive-RAG:**
**Query Type:** C ( Complex Query, Synchronous Retrieval )
**Step1:** retrieval and answer "when did Lionel Cranfield, 3rd Earl of Middlesex succeed his brother James as Earl of Middlesex?"
**Step2:** retrieval and answer "who is Lionel Cranfield, 3rd Earl of Middlesex's wife?"
**LLM Prediction:** 1651 && Rachael
**retrieval steps:** 2

**HANRAG:**
**Query Type:** D ( Compound Query, Asynchronous Retrieval )
**Step1:** retrieval and answer "when did Lionel Cranfield, 3rd Earl of Middlesex succeed his brother James as Earl of Middlesex?"
**Step1:** retrieval and answer "who is Lionel Cranfield, 3rd Earl of Middlesex's wife?"
**LLM Prediction:** 1651 && Rachael
**retrieval steps:** 1

---

Table 8: Comparison between HANRAG and Adaptive-RAG for Compound Query. Adaptive-RAG classifies the original query as a complex query and employs a Asynchronous retrieval approach. It first retrieves and generates the answer to the first sub-question, and then applies the same process to obtain the answer to the second sub-question. This results in a total of 2 retrieval steps for the original compound query. In contrast, HANRAG classifies the original query as a compound query and adopts an asynchronous retrieval approach. It simultaneously retrieves information for both sub-queries, leveraging a space-for-time trade-off to reduce the overall time required for the LLM to generate the final answer. **To simplify the presentation, only 3 documents are displayed here.**

**Query:** What is the Danish Football Union an instance of?
**Groundtruth:** International Federation of Association Football

**Adaptive-RAG:**
**Query Type:** C ( Complex Query, Asynchronous retrieval )
**Step1:** retrieval for "What does the acronym of the organization Danish Football Union is part of stand for?"
**Retrieved Doc1:** Lyngby Boldklub: Lyngby Boldklub () is a professional Danish football club founded in 1921. It is based at Lyngby Stadion in Kongens Lyngby, Denmark. From 1994 to 2001 the club was known as Lyngby FC. The club has won the Danish championship twice (1983 and 1992) and the Danish Cup three times (1984, 1985 and 1990).
**Retrieved Doc2:** Peter Møller: Peter Møller-Nielsen (born 23 March 1972) is ....
**Answer round 1:** Football association
**Step2:** retrieval for "What does the acronym of the organization Danish Football Union is part of stand for?"
**Retrieved Doc1:** Denmark national futsal team: The Denmark national futsal team is controlled by the Danish Football Association, the governing body for futsal in Denmark and represents the country in international futsal competitions, such as the FIFA Futsal World Cup and UEFA Futsal Championship.
**Retrieved Doc2:** Lyngby Boldklub: Lyngby Boldklub () is,...
**Retrieved Doc3:** Hobro IK: Hobro IK is,...
**Final Answer:** UEFA

**HANRAG:**
**Query Type:** C ( Complex Query, Asynchronous Retrieval )
**Step1:** retrieval for "What is the Danish Football Union an instance of?"
**Retrieved Doc1:** Denmark national futsal team: The Denmark national futsal team is controlled by the Danish Football Association, the governing body for futsal in Denmark and represents the country in international futsal competitions, such as the FIFA Futsal World Cup and UEFA Futsal Championship.
**Retrieved Doc2:** Lyngby Boldklub: Lyngby Boldklub, ...
**Retrieved Doc3:** ...
**Answer round 1:** FIFA
**Step2:** retrieval for "What does the FIFA stand for?"
**Retrieved Doc1:** Swiss are fans of football and the national team is nicknamed the 'Nati'. The headquarters of the sport's governing body, the International Federation of Association Football (FIFA), is located in Zürich,... is located in Switzerland and is named the Ottmar Hitzfeld Stadium.
**Retrieved Doc2:** Denmark national futsal team: The Denmark national futsal team is controlled by the Danish Football Association,...
**Retrieved Doc3:** ...
**Final Answer:** International Federation of Association Football

Table 9: Comparison between HANRAG and Adaptive-RAG for Complex Query

## H PROMPT LIST

You are an expert in English and can see through the essence of any English sentence.

Your current task is to fully understand and analyze the problem I gave you, and decompose the problem to obtain several sub-problems that constitute the problem. You need to follow the following rules:

Rule 1: Your output must be in json format, which contains only 2 keys. The first key is "thought" which represents your analysis and thinking process, and the second key is "decomposition" which represents the list of sub-problems after decomposition;

Rule 2: The question I give you may be the simplest one, that is, it only consists of one question and cannot be decomposed into other sub-problems. In this case, you only need to return the original question to me;

Now I will give you some examples to help you better understand and perform this task:

Example-1:

Query: Who was the first president of the United States?

Answer: {"thought": "This question is a very direct and simple question. There is no need to decompose it. It itself consists of only one sub-problem", "decomposition": ["Who was the first president of the United States?"]}

Example-2:

Query: What honors has Liu Xiang won and when did he retire?

Answer: {"thought": "This question consists of two sub-questions. On the one hand, it asks about the honors Liu Xiang has won, and on the other hand, it asks about the time when Liu Xiang retired, so the original question can be decomposed into two sub-questions.", "decomposition": ["What honors has Liu Xiang won?", "When did Liu Xiang retire?"]}

Example-3:

Query: What departments are there in Mayo Clinic, and which are the most famous ones?

Answer: {"thought": "This question consists of two sub-questions. On the one hand, it asks about the department composition of Mayo Clinic, and on the other hand, it asks about which are the most famous departments of Mayo Clinic, so the original question can be decomposed into two sub-questions.", "decomposition": ["What departments are there in Mayo Clinic?", "What are the most famous departments of Mayo Clinic?"]}

Now I will give you a question, please split it strictly according to the above rules and examples:

Query: <your_query>

Answer:

Table 10: Prompt for decomposer train and inference

You are an expert who is proficient in English and can see through the essence of any English sentence.

Your current task is to fully understand and analyze the question I gave you, and tell me whether it is a super simple common sense question, a simple single-step search question, a compound question, or a complex logical reasoning question. I will now give you the definitions of these types of questions:

1. Straightforward question, which means that this question does not require external knowledge to be queried, and the information you know is enough to answer the question;

2. Single-step question, which means that the information you know cannot answer this question, and you need to use some external knowledge, such as searching the Internet, asking experts, etc. to answer it, but you only need to use external knowledge once;

3. Compound question, which means that this question is composed of multiple sub-questions, but these sub-questions are not related, or the correlation is relatively small, and no complex logical reasoning is required, but the information you know cannot answer the question, and it needs to be broken down into several sub-questions and then answered with the help of an external knowledge base;

3. Complex question, which means that this question is composed of multiple sub-questions through complex logical nesting. There is a very strong logical relationship between these sub-questions. After decomposition, you still need to get the answer to a sub-question before you can continue to answer other sub-questions. That is, the answer to sub-question 1 is the prerequisite for sub-question 2.

Your output needs to follow the following rules:

Rule 1: You need to fully understand and analyze the given query and give your answer;

Rule 2: You only need to give the type of question, and other content is prohibited.

Now I will give you some examples to help you better understand and perform this task:

Example-1:
Query: Who is the first President of America?
Answer: straightforward question
Example-2:
Query: Which company acquired Intime Department Store?
Answer: single-step question
Example-3:
Query: What honors did Yao Ming win in the NBA? When did he retire from the NBA?
Answer: compound question
Example-4:
Query: What city is the person who broadened the doctrine of philosophy of language from?
Answer: complex question
Example-5:
Query: What is the scientific classification of conch shells, and what are the common uses of conch shells in various cultures?
Answer: compound question
Example-6:
Query: In which country was Einstein born?
Answer: straightforward question
Example-7:
Query: Who is Colin Kaepernick and what is his preferred nickname?
Answer: complex question
Example-8:
Query: Where is Pan Jianwei's ancestral home?
Answer: single-step question
Now I will give you a query. Please fully understand it and output it according to the above example and strictly abide by the rules:
Query: <your_query>
Answer:

Table 11: Prompt for revelator inference

You are a linguist, proficient in various literary works, and can easily see through the essence of any English sentence.

I want to answer a question, which may be a simple question or a very complex question that requires multiple steps of reasoning to answer. For simple questions, I only need to search once in the search engine to get the answer; for complex questions, I need to solve them step by step. First, I need to refine the first seed question that needs to be answered in the complex question, and then I can further answer the next step of the complex question after answering it. What you need to do is to help me find the seed question in the question I gave.

I will give you two aspects of content. The first is the complex problem mentioned above. The second is some solution steps I got after thinking and disassembling, including multiple seed questions refined from several steps of reasoning, and the answers to these seed questions. Given these two aspects of content, please help me refine the first seed question that needs to be answered in the complex problem. You need to abide by the following rules:

Rule 1: If the problem given to you is a complex problem, then what you need to do is to refer to the thinking process I have completed and help me refine the seed question that needs to be answered next to this complex problem, that is, the first sub-question that must be answered first to answer this problem;

Rule 2: If the problem given to you is a simple single-step problem, then you only need to output the original problem intact;

Rule 3: You must not output any other irrelevant content, which is very important;

Rule 4: The several parts of content I give you are "Question" for complex problems, "Thought" for completed thinking process, if its content is "nothing", it means that there is no completed thinking process, and "Output" for the content you need to output.

Now I will give you some examples to help you better understand this task:
Example-1:
Question: Where was the director of film Eisenstein In Guanajuato born?
Thought: "'nothing"'
Output: Who is the director of the film Eisenstein In Guanajuato?
Example-2:
Question: Who is the first President of America?
Thought: "'nothing"'
Output: Who is the first President of America?
Example-3:
Question: Who is the father-in-law of Queen Hyojeong?
Thought:
"'
**seed query-1**: Who is the husband of Queen Hyojeong?
**answer-1**: Heonjong of Joseon
"'
Output: Who is the father of Heonjong of Joseon?

Now I will give you a question. You should output according to the above rules and examples. Do not output any irrelevant content:
Question: <your_query>
Thought:
"'
<your_thought>
"'
Output:

Table 12: Prompt for refiner train and inference

You are a linguist proficient in various literary works.

Your current task is to determine whether the document and the question I provide are related. I will give you a document and a question. This question is a real user's inquiry, and the document is content I have retrieved. The document may or may not be related to the question, so you need to make a judgment.

You must follow these rules:

Rule 1: If the doc is related to question, you must output true; if not, output false.

Rule 2: A very important principle for determining relevance is that if the content of the document can be used to answer the question, whether it directly answers the question or merely serves as a reference to answer the question, it should be considered relevant.

Rule 3: You can only output true or false, and nothing else.

Now I will provide you with some examples to help you better understand and perform this task:

Example-1:

Question: Who was the first president of the United States?

Doc: George Washington (February 22, 1732 – December 14, 1799) was the first president of the United States, serving from 1789 to 1797. As commander of the Continental Army, Washington led Patriot forces to victory in the American Revolutionary War against the British Empire. He has become commonly known as the "Father of His Country" for his role in American independence.

Answer: true

Example-2:

Question: What honors has Liu Xiang received, and when did he retire?

Doc: Liu Xiang (born July 13, 1983), born in Shanghai, with ancestral roots in Xihe Village, Dafeng, Yancheng, Jiangsu, is a Chinese male athlete. He won one Olympic gold medal, six World Championship medals, and three Asian Games gold medals. He is a two-time world champion and held the 110m hurdles world record for 23 months, which still stands as the Olympic record.

Answer: true

Example-3:

Question: Who founded the Mayo Clinic?

Doc: The Mayo Clinic is a medical institution located in Rochester, Minnesota, USA, established in 1864. It has branches in Jacksonville, Florida, and Scottsdale, Arizona, as well as smaller clinics and hospitals in Minnesota, Iowa, and Wisconsin. It is consistently ranked as the best hospital in the world by major authoritative reports.

Answer: false

Now, I will provide you with a question and a document. Please strictly follow the above rules and examples to analyze and output the answer:

Question: <your_query >

Doc: <your_doc >

Answer:

Table 13: Prompt for relevance discriminator train and inference

You are an expert who is proficient in various fields.

Your current task is to answer the questions I give you based on the documents I give you. The questions are real questions from users, and the documents are some information related to the questions that you have retrieved. You must deliver your predictions in the most concise language. For example, if the answer is a person, just output their name; if the answer is a specific time, simply output the time point; if the answer is "yes" or "no" just output "yes" or "no".

Now let me give you some examples to help you better understand this task:

Example-1:

Question: Which year did Liu Xiang retire?

Doc1: "'Liu Xiang (born July 13, 1983), born in Shanghai, with ancestral roots in Xihe Village, Dafeng, Yancheng, Jiangsu, is a Chinese male athlete. He won one Olympic gold medal, six World Championship medals, and three Asian Games gold medals. He is a two-time world champion and held the 110m hurdles world record for 23 months, which still stands as the Olympic record.'"

Doc2: "'The Mayo Clinic is a medical institution located in Rochester, Minnesota, USA, established in 1864. It has branches in Jacksonville, Florida, and Scottsdale, Arizona, as well as smaller clinics and hospitals in Minnesota, Iowa, and Wisconsin. It is consistently ranked as the best hospital in the world by major authoritative reports.'"

Doc3: "'On April 7, In 2015, Liu announced his retirement in a statement posted to his Sina Weibo. He had not competed since the 2012 Olympic race.'"

Answer: 2015

Example-2:

Question: Did Nanjing University found in 1958?

Doc1: "'Nanjing University, located in the ancient capital of China - Nanjing, is one of the oldest and most prestigious institutions of higher learning in the country. Founded in 1902 as Sanjiang Normal School, it has since evolved through various transformations to become the comprehensive university we know today. Renowned for its strong emphasis on academic research and teaching excellence, Nanjing University offers a wide range of disciplines including humanities, social sciences, natural sciences, engineering, and medicine.'"

Doc2: "'The University of Science and Technology of China (USTC), founded in 1958 in Beijing and later relocated to Hefei, Anhui Province, is a premier institution dedicated to fostering academic excellence and innovation. USTC is particularly renowned for its strong emphasis on science and technology education and research. As one of the key universities under the national Double First-Class University Plan, it has established itself as a leader in various scientific disciplines including physics, chemistry, life sciences, engineering, and information technology.'"

Doc3: "'Nanjing University stands out for its exceptional academic programs across various fields, with several disciplines earning national and international acclaim. The university's Astronomy department is particularly noteworthy, boasting a rich history and pioneering research in astrophysics, cosmology, and radio astronomy. Additionally, the Earth Sciences division, including Geology and related fields, is highly regarded for its comprehensive studies in paleontology, stratigraphy, and tectonic geology.'"

Doc4: "'The University of Science and Technology of China (USTC), located in Hefei, Anhui Province, is renowned for its strong emphasis on science and technology education. Established in 1958, USTC has been a pioneer in fostering innovation and cutting-edge research in various scientific fields. On the other hand, the University of Chinese Academy of Sciences (UCAS), with its main campus in Beijing, focuses on graduate education and high-level scientific research. UCAS, established much later in 2012, collaborates closely with the Chinese Academy of Sciences, offering students unique opportunities to engage in advanced research projects under the guidance of leading scientists.'"

Now I will give you a question and several documents that you need to fully understand before giving the answer, You only need to output the answer, do not output your thought process or other irrelevant information:

Question: <your_query>

<your_doc_list>

Answer:

Table 14: Prompt for generator train and inference

You are a middle school English teacher.

Your task is to refine a question based on the topic and document I give you, and find the answer to this question from the document. This task is equivalent to building an exam question based on the given document and around the topic.

You need to follow the following rules:

Rule 1: Your output must be in JSON format, containing two keys. The first one is "Question" which means the question you asked based on the given document around the given topic, and the second key is "Answer", which means the answer to the question that can be found directly from the document.

Rule 2: The question you ask must be very simple, and the answer to this question must be answered in a few words, because you are giving exam questions to low-grade junior high school students, and their English level can only find the answer to the question in the document.

Rule 3: The question you ask must be able to find the answer directly from the document, and the answer you give must be a simple entity containing only a few words, because this will be used as the correct answer to the exam question to calculate the student's score.

I'll give you some examples now:

Example1:

Title: Liu Xiang

Doc: Liu Xiang is a legendary Chinese hurdler, widely recognized as one of the greatest athletes in Chinese sports history. He was born on July 13, 1983, in Shanghai. Liu Xiang rose to international fame in 2004 when he won the gold medal in the 110-meter hurdles at the Athens Olympics, becoming the first Chinese male athlete to win an Olympic gold medal in track and field. His victory was historic as he equaled the world record of 12.91 seconds, set by Colin Jackson.

Output: {"Question": "Which year was Liu Xiang born?", "Answer": "1983"}

Example2:

Title: Yao Ming

Doc: Yao Ming, born on September 12, 1980, is a retired Chinese professional basketball player who played as a center. Standing at 7 feet 6 inches (2.29 meters) tall, he was one of the tallest players in the NBA during his career and became a cultural icon both in China and internationally. Drafted by the Houston Rockets as the first overall pick in the 2002 NBA draft, Yao spent his entire NBA career with the Rockets from 2002 to 2011.

Output: {"Question": "What sports did Yao Ming play?", "Answer": "basketball"}

Now you need to generate according to the above example, you only need to output one question, please do not output any other irrelevant content:

Title: <your_title >

Doc: <your_doc >

Output:

Table 15: Prompt for single query construction from wiki corpus

I will give you several simple questions. Please combine them into a compound question. Since these questions are all about a certain entity, you need to follow the following rules:

Rule 1: You need to fully understand the given questions and combine them perfectly;

Rule 2: If the given questions cannot be combined, you only need to output "no";

Rule 3: If they can be combined, the combined question must be a compound question, that is, this question must be about a certain entity and ask about several different aspects of the entity.

Now let me give you some examples for your reference:

Example1:

Simple Question1: "'When was Arthur's Magazine first published? "'

Simple Question2: "'What is the main focus of Arthur's Magazine content? "'

Compound Question: "'When was Arthur's Magazine first published, and what is the main focus of its content? "'

Example2:

Simple Question1: "'What frequency does KMBZ-FM broadcast on? "'

Simple Question2: "'What music did KMBZ-FM play in 1975? "'

Simple Question3: "'What was the share of KMBZ in the Kansas City Arbitron ratings report in February 2011? "'

Compound Question: "'What is the broadcasting frequency of KMBZ-FM, what type of music did it play in 1975, and what was its share in the Kansas City Arbitron ratings report in February 2011? "'

Now I will give you these simple questions. You must strictly follow the above rules to output them. It is strictly forbidden to output any other irrelevant content:

<simple_questions >

Compound Question:

Table 16: Prompt for compound queries construction