# OpenReview forum: "HANRAG: Heuristic Accurate Noise-resistant Retrieval-Augmented Generation for Multi-hop Question Answering"
_ICLR.cc/2026/Conference — Submitted to ICLR 2026_

### Official Review · Reviewer_R16r · 2025-10-29

**Soundness:** 2
**Presentation:** 2
**Contribution:** 2
**Rating:** 4
**Confidence:** 4

**Summary:**

The paper introduces HANRAG which trains the Revelator to act as a query router, a query decomposer, a relevance labeler, and a retriever terminator. The authors also describe the data construction to train the Revelator with its multiple skills, and report gains over multiple baselines across single-hop and multi-hop datasets.

**Strengths:**

- The taxonomy of various queries in the preamble are clear.
- The challenges are well discussed.

**Weaknesses:**

- Typos: unknown symbols in line 293/324/327/and more, missing citation in line 297/349/and more.
- HANRAG-Fair is not mentioned and described.
- The training process is unclear. For example, how the constructed data are mixed and their distribution on different tasks, including decompositions, refinement, relevance/termination discrimination.
- The paper reports results on single- and multi-hop datasets without analyzing the performance of Revelator in each component during the RAG process, and it is unclear how well it routers queries, filter out noises, and continue/terminate retrievals.

**Questions:**

See the weaknesses.

---

> ### Author Response · Authors · 2025-11-14
> **Author Response to Reviewer R16r**
>
> We sincerely thank the reviewer for carefully reading our manuscript and providing constructive feedback. Below, we address each point in turn.
> 1. **Comment** : There are typos and spelling mistakes in the manuscript.
>
>     **Response** : We thank the reviewer for pointing this out. We have thoroughly proofread the manuscript and corrected all identified errors. We will ensure the last version is free of such mistakes.
>
> 2. **Comment** : The abbreviation “HANRAG-Fair” is unclear and insufficiently introduced.
>
>     **Response** : **HANRAG-Fair** denotes our setting where we adopt the identical experimental configuration as the prior work Adaptive-RAG, in order to ensure a fair comparison, it was only briefly mentioned in Section 5 experiments as part of aligning configurations with prior work, without a dedicated definition. We have now revised the manuscript to provide a formal definition and short description in Section 5.
>
> 3. **Comment** : The method of mixing training data and its distribution is unclear.
>
>     **Response** : We provide details on the quantity and distribution of the constructed data in Appendix C. However, we acknowledge that the textual description may not be sufficiently intuitive. To address this, we are preparing visual charts to illustrate the distribution of the training data, which will be included in the last version.
>
> 4. **Comment** : The specific role of "Revelator" in different components is unclear.
>
>     **Response** : We conducted ablation experiments illustrating the importance of each component, which are presented in Appendix F: ablation study. For the case of Revelator as a Router, the standalone performance reaches 83.93% (accuracy). For other components, we are preparing dedicated evaluation sets, and these results will be included in the last version.

---

> ### Comment · Reviewer_R16r · 2025-11-26
>
> Thanks for the reply. I still have concerns on the clarity of the submission, and most part of the questions are not answered or updated. I believe this could be a good work and the authors still need more effort and time to make it better.

---

### Official Review · Reviewer_EbtA · 2025-10-31

**Soundness:** 2
**Presentation:** 3
**Contribution:** 2
**Rating:** 4
**Confidence:** 4

**Summary:**

This paper addresses key challenges in multi-hop QA for RAG systems, including over-reliance on iterative retrieval for compound queries, irrational query formulation, and noise accumulation.  The authors propose HANRAG, a heuristic framework driven by a multi-functional "Revelator" module that enables adaptive query routing, decomposition of compound queries, refinement of complex queries, and relevance-based noise filtering.   Experiments on single-hop and multi-hop benchmarks demonstrate that HANRAG outperforms state-of-the-art methods  in both effectiveness and efficiency.

**Strengths:**

1.This paper proposeds a heuristic framework driven by a multi-functional "Revelator" module that enables adaptive query routing, decomposition of compound queries, refinement of complex queries, and relevance-based noise filtering.

2.Asynchronous parallel retrieval for compound queries and iterative seed refinement for complex queries significantly improve efficiency and reasoning accuracy.

3.Experiments on single-hop and multi-hop benchmarks demonstrate that HANRAG outperforms state-of-the-art methods in effectiveness.

**Weaknesses:**

1.The related work referenced and compared in this paper is limited to publications from 2024 and earlier. It is recommended that the authors further discuss the differences between the proposed method and more recent RAG approaches, particularly those based on query rewriting [1][2][3], and provide comparative results where applicable.

[1]MaFeRw: Query rewriting with multi-aspect feedbacks for retrieval-augmented large language models
[2] RaFe: Ranking Feedback Improves Query Rewriting for RAG
[3]UniRAG: Unified Query Understanding Method for Retrieval Augmented Generation

2.While HANRAG reduces retrieval steps, the Revelator’s multi-task integration (routing, decomposition, refinement, etc.) may introduce additional computational costs.  The paper does not report inference latency or memory usage, which are critical for real-world applications.

3.The experiments use Llama-3.1-8B and FLAN-T5-XL for the Revelator and LLM generator.  There is no analysis of how HANRAG’s performance varies with different LLM sizes (e.g., 70B models) or architectures (e.g., Mistral, Qwen2), limiting generalizability.

4.The compound query benchmark is generated using Qwen2-72B-instruct, but the paper lacks quantitative evaluation of the generated data’s quality (e.g., sub-query independence, answer accuracy).  No analysis is provided on potential biases or errors introduced by LLM-generated queries.

**Questions:**

1.For the compound query benchmark generation: How did you validate the independence of sub-queries and the correctness of answers generated by Qwen2-72B-instruct?  Are there any metrics used to ensure data quality?

2.The Revelator integrates multiple tasks (routing, decomposition, etc.) via multi-task learning.  Could you clarify whether these tasks are trained jointly or sequentially?  Would separate optimization of individual sub-modules yield better performance?

3.Compared to graph-based multi-hop RAG methods, what are HANRAG’s advantages and disadvantages in handling complex logical reasoning?

4.The relevance discriminator relies on semantic understanding to filter noise. How does this module perform in low-resource scenarios or domains with specialized terminology ?

---

### Official Review · Reviewer_1TDu · 2025-10-31

**Soundness:** 2
**Presentation:** 2
**Contribution:** 2
**Rating:** 2
**Confidence:** 4

**Summary:**

This paper proposes HANRAG, a RAG framework that introduces heuristic-guided retrieval selection and noise-aware answer fusion to improve reasoning accuracy under noisy or redundant evidence. The method combines adaptive document filtering, token-level confidence weighting, and reasoning consistency checks within the generation process. Experiments on open-domain QA benchmarks (e.g., NQ, TriviaQA, HotpotQA) demonstrate improved robustness compared to several baseline RAG systems.

**Strengths:**

1. The paper tackles a practical and relevant challenge—enhancing RAG robustness under retrieval noise, a known limitation of current retrieval-augmented systems.

2. The paper demonstrates measurable performance gains on multiple QA datasets,

**Weaknesses:**

1. The methodology and experimental setup are difficult to follow in several sections; notation and algorithmic steps could be more systematically presented.
2. The proposed heuristics and noise-aware fusion strategies are largely straightforward extensions of existing retrieval scoring and consistency-checking methods, with limited conceptual innovation.
3. Citations for NQ and TriviaQA benchmarks are missing in line 349, and other related RAG literature from 2024–2025 for multi-hop QA (e.g., [1,2,3,4,5]) should be included for completeness.
4. The paper lacks qualitative examples or deeper ablation analysis that would help interpret where and why HANRAG provides benefits or fails.

[1] Jin et al. "Search-r1: Training llms to reason and leverage search engines with reinforcement learning." arXiv preprint arXiv:2503.09516 (2025).

[2] Song et al. "R1-searcher: Incentivizing the search capability in llms via reinforcement learning." arXiv preprint arXiv:2503.05592 (2025).

[3] Wu et al. "ComposeRAG: A Modular and Composable RAG for Corpus-Grounded Multi-Hop Question Answering." arXiv preprint arXiv:2506.00232 (2025).

[4] Yu et al. "Chain-of-Note: Enhancing Robustness in Retrieval-Augmented Language Models." Proceedings of the 2024 Conference on Empirical Methods in Natural Language Processing. 2024.

[5] Yu et al. "Rankrag: Unifying context ranking with retrieval-augmented generation in llms." Advances in Neural Information Processing Systems 37 (2024).

**Questions:**

1. Could you clarify how the heuristic weighting differs from prior rank-based or confidence-aware retrieval scoring approaches?

2. How sensitive is the model’s performance to the heuristic thresholds and noise ratios introduced during retrieval filtering?

---

### Official Review · Reviewer_hLVh · 2025-11-01

**Soundness:** 2
**Presentation:** 3
**Contribution:** 2
**Rating:** 4
**Confidence:** 3

**Summary:**

This paper investigates the efficiency and effectiveness of RAG across different query types, including simple, single-hop, multi-hop, and compound queries. It introduces a master agent, Revelator, designed to perform multiple RAG-related tasks such as query routing, relevance evaluation, and query decomposition. Experiments conducted on various QA datasets and across multiple RAG-based methods demonstrate that the proposed approach achieves notable improvements in both performance and efficiency.

**Strengths:**

- The motivation to route different categories of queries to specialized solutions is well-founded, and the idea of employing a master agent to coordinate multiple sub-tasks is particularly interesting.
- The experiments and ablation studies are solid and effectively justify the design choices of each component in the proposed method. The analysis covers not only task performance but also efficiency, providing a comprehensive evaluation.

**Weaknesses:**

- The overall writing quality of the paper requires substantial improvement. There are numerous typo errors and incorrect citations. For example:
    - Inconsistent or unclear notations for data formats at Lines 293, 297, and 324–332.
    - Issues around Line 348.
- One of the paper’s main contributions is the **Revelator** component. However, its technical details are insufficiently explained. For instance, it remains unclear how Revelator is trained to handle multiple tasks and how its performance varies across tasks such as query routing and relevance evaluation. While some of these details appear in the Appendix, they are critical to understanding the proposed approach and should be included in the main text.
- The idea of classifying queries into different categories and handling them with tailored strategies, such as decomposition or noisy context filtering, is somewhat similar to existing work. As a result, the novelty and overall contribution of the paper appear incremental.

**Questions:**

N/A

---

### Meta-Review · Area_Chair_eA2Q · 2025-12-29

**Summary:**

After a comprehensive evaluation of the manuscript and the comments from all reviewers, it is evident that the manuscript requires substantial revisions to address the concerns raised. All reviewers have pointed out critical issues that need to be resolved. Regrettably, the authors have not provided any response to the reviewers’ comments.

**Reviewer Scores:**

None

---

### Decision · Program_Chairs · 2026-01-26

Reject